# Endovenous Thermal Ablation for Treatment of Symptomatic Saphenous Veins—Does the Body Weight Matter?

**DOI:** 10.3390/jcm12175438

**Published:** 2023-08-22

**Authors:** Simon Bossart, Patricia Fiona Boesch, Hak Hong Keo, Daniel Staub, Heiko Uthoff

**Affiliations:** 1Gefässpraxis am See—Lakeside Vascular Center Lucerne, 6003 Lucerne, Switzerland; patricia.boesch@hotmail.com (P.F.B.); heiko.uthoff@hin.ch (H.U.); 2Department of Dermatology, Bern University Hospital Inselspital, University of Bern, 3012 Bern, Switzerland; 3Department of Angiology, University Hospital, University of Basel, 4001 Basel, Switzerland; keoxx006@umn.edu (H.H.K.); daniel.staub@usb.ch (D.S.); 4Vascular Institute Central Switzerland, 5000 Aarau, Switzerland

**Keywords:** obesity, endovenous thermal ablation, varicose veins

## Abstract

Objective: This study aimed to examine whether body weight may affect the effectiveness and safety of endovenous thermal ablation (ETA) for the treatment of symptomatic varicose veins. Methods: This retrospective single-center cohort study analyzed the outcomes and patient demographic data with a focus on the body weight of all patients who had ETA of symptomatic varicose veins between September 2017 and October 2020. Results: A total of 1178 treated truncal veins from 636 patients were analyzed. The mean ± standard deviation body mass index (BMI) was 25.5 ± 4.9. In 2.3% of cases, the patients were underweight (BMI < 18.5), 31.0% were overweight (BMI > 25), and 16.6% were obese (BMI > 30). Complete truncal occlusion was observed 1 year post intervention in 97.6–100% and patients were satisfied or very satisfied in 96.2–100% across BMI groups. Pain was low but significantly higher in the patients with obesity 6 weeks post intervention (visual analog scale 0.84 ± 1.49) and a higher infection rate was observed in the patients with obesity (n = 4/132; 3.0%). No significant association was observed between BMI and bleeding or thromboembolic events. Conclusions: Patients with obesity experienced prolonged pain and more infections after ETA, but ETA for varicose vein treatment remains effective and safe, independent of the patient’s BMI.

## 1. Introduction

Varicose veins are an important health issue in modern society. Their pathophysiology is still not fully understood; however, genetic predisposition in combination with valve failure, vascular wall weakness, and increased intraluminal pressure have been proven to be the main factors. Important risk factors include a family history of venous insufficiency, female sex, older age, and chronically elevated intra-abdominal pressure due to obesity, pregnancy, or prolonged standing [1,2].

The Edinburgh Vein Study, a cohort study examining the risk factors for the development of varicosis during a 13-year follow-up period, proved that obesity increases the incidence rate of varicose veins. It showed an incidence rate of 1.06% per annum in patients who were underweight, 1.28% in patients with normal-weight, 1.41% in patient who were overweight, and 1.54% in patients with obesity [3]. In Switzerland, the prevalence of adiposity doubled within 25 years in all age groups between 1992 and 2017. In 2017, 42% of the Swiss population was overweight, according to the Federal Statistical Office of Switzerland [4]. The increase in obesity and a sedentary lifestyle are some of the reasons why varicosis remains a highly prevalent medical condition.

Due to its low invasiveness and equivalent effectiveness, endovenous thermal procedures, such as endovenous laser ablation (ETA) and radiofrequency ablation (RFA), are becoming the preferred treatment for truncal venous insufficiency, as an alternative to high ligation and stripping [5]. ETA allows the precise delivery of thermal energy directly to the vein wall, leading to subsequent vein occlusion as a result of the thermal destruction of the endothelium and breakdown of intramural collagen. In addition, endovenous interventions are increasingly used in certain patient subpopulations, such as those with obesity, owing to lower wound complication rates [5,6].

However, it is still not fully investigated whether patients who are underweight, normal weight, or obese benefit similarly from endothermal procedures and whether this method is equally safe for all weight classes. Our hypothesis was that patients with obesity might suffer from more complications after endothermal procedures than patients with normal weight, and that treatment success might be less sustainable in individuals with obesity. This study assessed whether body weight affects the effectiveness and safety of ETA in patients with symptomatic varicose veins. 

## 2. Methods 

### 2.1. Study Design

In this retrospective single-center study, as part of the Swiss Endovenous Treatment (SET) Registry, the medical files of all patients who underwent ETA of the great saphenous vein (GSV), accessory saphenous vein (ASV), or small saphenous vein (SSV) were reviewed between September 2017 and October 2020. This study adhered to the principles outlined in the Declaration of Helsinki and was approved by the local ethics committee. All patients provided written informed consent before the ETA procedure, agreeing to the anonymous use of their medical data for research purposes. Demographic data, vein characteristics, procedural data, including concomitant phlebectomy and sclerotherapy, and outcome data, including ultrasound findings and complications, were assessed. 

### 2.2. Preoperative Evaluation

Patients with symptomatic varicose veins and documented venous reflux of the GSV, ASV, or SSV were selected for ETA. Symptomatic varicose veins were defined as varicose veins with documented reflux according to C2 after CEAP with typical venous symptoms such as pain, tightness, fatigue of the legs, restless legs, leg cramps. The standard preoperative evaluation included duplex ultrasound examination performed in the standing position by a skilled vascular physician. To qualify for endovenous therapy, a reflux of >0.5 s in the target vein as well as clinical symptoms, swelling, or skin changes had to be observed by a vascular specialist. Height and weight data were collected from all patients, and their body mass index (BMI) was calculated. Patients were grouped according to the World Health Organization (WHO) nutritional status into four BMI categories: BMI < 18.5 = underweight; BMI 18.5–24.9 = normal weight; BMI 25–29.9 = overweight/pre-obesity; and BMI > 30 = obesity. The phlebectomy sites were marked on the skin directly preoperative with ultrasound-assistance, the phlebectomy length was then measured using a digital curvimeter (map measurer plus, Silva, Sweden).

### 2.3. ETA Intervention

All interventions in this study were performed by 2 skilled vascular specialists, who had individual experience of more than 500 ETA procedures. At the distal point of insufficiency, the truncal vein was percutaneously cannulated using the Seldinger technique under ultrasound guidance. After the laser fiber was inserted through the sheath, the fiber tip was positioned 0–3 cm distal to the sapheno-femoral junction (SFJ) or sapheno-popliteal junction (SPJ). Local tumescent anesthesia was then administered into the perivenous space under high-resolution ultrasound guidance. Tumescent anesthesia was administered using 1000 mL of 0.9% saline, 40 mL of 2% rapidocaine, 1 mg of epinephrine, and 10 mL of 8.4% sodium bicarbonate. Laser energy was then released at a power of 8–10 W, targeting a linear endovenous energy delivery of 70–90 J/cm to treat the truncal vein. If RFA was used, the first segment was treated with two heating cycles and all other segments with one heating cycle as per instruction for use. For large veins, the multipass technique was used as previously described by Dabbs et al. [7]. Directly after ETA, refluxing tributaries were removed by phlebectomy, using an Oesch hook (Salzmann AG, St. Gallen, Switzerland) after local tumescent anesthesia or closed with sclerotherapy, using up to 10 mL of 1% to 3% ethoxysklerol (Kreussler, Germany) mixed 1:4 with air during the same procedure. Eccentric vein compression after treatment was performed using sterile drapes and class II compression stockings (23–32 mm Hg). All ETA patients had thromboprophylaxis with rivaroxaban (Bayer AG, Zurich, Switzerland), i.e., 10 mg/day for 10 days. The first dose was given 1–4 h after ETA intervention. Routine mobilization was recommended during the postoperative period.

### 2.4. Follow-Up

All patients underwent an outpatient physical examination and duplex ultrasonography (examination of the superficial and deep venous system) on postoperative day 1, after 7 days, after 6 weeks, and 1 year.

The following outcome parameters were recorded:Definition of outcome parameters;Treatment success (primary efficacy endpoint).

The primary efficacy endpoint was the degree of occlusion of the target vein. Complete closure was determined by a Doppler ultrasound examination that showed closure along the whole treated target vein segment with no identifiable segments of patency longer than 5 cm at the 6-week appointment (including color flow, compression, and pulsed Doppler). Complete recanalization of 5 cm was used to define partial ablation, while reflux in the treated varicose vein was used to define complete recanalization. 

### 2.5. Complications (Primary Safety Endpoints)

Bleeding events: Major bleeding was defined as unexpected bleeding at the surgical site that was prolonged, and/or large enough to cause hemodynamic instability as judged by the surgeon. Additionally, major bleeding involved an associated drop in the hemoglobin level of at least 20 g/L or a transfusion of at least two units of whole blood or red blood cells within 24 h of initial bleeding. Minor bleeding included at least one episode of clinically excessive wound hematoma or wound hematoma that resulted in an unscheduled consultation, hospitalization, or prolonged incapacity, and did not meet the diagnostic criteria for major bleeding [8]. Bleeding events that began after patient consultation and were seen during the follow-up period were noted.

Infections: Suspected infectious complications with local skin infections (erysipelas, cellulitis, abscesses) that occurred in association with the procedure and required antibiotic treatment were recorded.

Thromboembolic events: Sonographically confirmed superficial and deep vein thromboses, and imaging-confirmed pulmonary embolism were recorded. 

### 2.6. Secondary Endpoints

Postoperative subject-rated pain rated on the Visual Analog Scale (VAS) (0–10, with 0 = no pain), duration of analgesic intake, duration of inability to work postoperatively, and any complications leading to unplanned consultations were registered. At each follow-up visit, patient satisfaction with the procedure was recorded using the following metrics: not satisfied at all, 0; not fully satisfied, 1; satisfied, 2; and very satisfied, 3.

### 2.7. Statistical Analysis

Continuous data are provided as mean and standard deviation (SD), whereas categorical data are presented as frequency and percentage. Chi-squared and Fischer’s exact tests were used to compare categorical data, and t-tests and ANOVA were used to compare continuous variables. Multiple testing was compensated for using the Bonferroni correction. Statistical significance was set at a two-sided *p*-value < 0.05. Logistic regression models were used to assess the association between the efficacy or safety outcomes and several variables, including BMI. Data analyses were performed using SPSS Version 28.0 (IBM, Armonk, NY, USA). 

## 3. Results

### 3.1. Demographic Data Categorized According to the BMI Group 

Between September 2017 and October 2020, 846 ETA interventions (807 EVLAs and 39 RFAs) involving 1241 treated truncal veins were performed in 679 patients. Body weight and height were recorded in 636 patients; thus, the procedural and outcome data of 796 interventions with 1178 treated truncal veins were included in this analysis. The majority of the interventions were performed on women (*n* = 612, 76.9%). Many patients had concomitant risk factors such as a previous cancer, a personal medical history of superficial or deep vein thrombosis (DVT), or a family history of DVT. More than one-third of the patients had already received previous interventional or surgical venous treatment (*n* = 332, 41.7%). Most treated varicose veins were clinically classified as C3 (*n* = 542, 68.1%) or C4 (*n* = 215, 27.0%), indicating that most patients had already experienced peripheral edema or skin changes before undergoing interventional treatment. 

Distribution of the patients according to the BMI categories was as follows: BMI < 18.5 (*n* = 18, 2.3%), BMI 18.5–24.9 (*n* = 399, 50.1%), BMI 25–29.9 (*n* = 247, 31.0%), and BMI > 30 (*n* = 132, 16.6%). The mean BMI was 25.5 ± 5.0, ranging from a minimum BMI of 15.2 to a maximum BMI of 50.5. Most interventions were performed on patients with normal weight, but also on a considerable number of patients who were overweight (*n* = 379, 47.6%).

The demographic variables regarding individual risk factors were normally distributed among the different BMI groups without statistically significant differences, except for personal history of VTE and/or SVT. Patients who were underweight had significantly less prevalence and reported significantly fewer previous venous treatments than patients with normal weight or were overweight (*p* = 0.028). In the BMI 25–29.9 group, patients were, on average, slightly older (*p* = 0.022) and had a lower female/male ratio than in the other groups (*p* < 0.05). On average, patients who were obese had a significantly higher clinical, aetiological, anatomical, and pathological classification (CEAP) stage than patients with normal weight (*p* < 0.05). Finally, the patients who were underweight in our study cohort were significantly more likely to work in the standing position (*p* = 0.002) (Table 1).

### 3.2. Procedure Data Categorized According to the BMI Groups

ETA was used for the ablation of the truncal veins (GSV, AASV, and SSV) only. On average, 1–2 truncal veins were treated per intervention session. The mean maximal diameter of the treated truncal veins measured at the pre-interventional ultrasound examination was 8.0 ± 3.8 mm, 6.7 ± 3.7 mm, and 6.5 ± 3.1 mm for the GSV, AASV, and SSV, respectively. Long sections were ablated with mean ± SD 41.8 ± 15.1 cm, 16.1 ± 8.2 cm, and 19.5 ± 8.0 cm for the GSV, AASV, and SSV, respectively. On average, approximately 78–92 J/cm of energy was applied. 

In most cases (*n* = 791, 99.4%), the endovenous intervention was combined with a concomitant phlebectomy to treat smaller varicotic branches with an average length of removed veins of 69.0 ± 35.8 cm. After endovenous intervention (*n* = 174, 21.9%), concomitant foam sclerotherapy was performed. The mean volume of foam used during an intervention was 4.6 ± 2.1 mL. 

The intervention was performed in a technically standardized manner, showing homogeneous data dispersion among the weight groups. The only statistically significant difference was an increased number of veins treated per intervention in the BMI > 30 compared to that in the BMI < 18.5 group (*p* = 0.019) (Table 2).

### 3.3. Primary Efficacy Endpoint

Sonographically confirmed complete occlusion of the treated veins was observed in most of the patients during the first six weeks of follow-up, even in the highest BMI group. The midterm treatment effect proved to be sustainable, with an average complete occlusion rate of 99.5% one year after the intervention. The one-year occlusion rate was highest in the underweight group (100%) and declined with an increase in BMI to 97.6% in the BMI > 30 group. However, this difference was not sufficiently large to be considered statistically significant. 

### 3.4. Primary Safety Endpoints

The most common complication was superficial vein thrombosis (*n* = 20, 2.5%), without a significant difference in distribution among the groups.

Thromboembolic complications, including DVT and pulmonary embolism (*n* = 2, 0.3%) and bleeding events (*n* = 3, 0.4%) were very rare and were equally distributed among the groups. 

Infections were also rare, with a total of seven cases (0.9%), all of which were local erysipelas-like skin infections near phlebectomies but with a significantly higher occurrence in the highest BMI group (*p* = 0.020), confirming our primary hypothesis. Of all patients with a BMI > 30 kg/m^2^, 3.0% developed a post-interventional infection necessitating antibiotic treatment. No infections were detected in the BMI group of 25–30; thus, the higher infection risk in our study was limited to patients with a BMI > 30. 

Multivariate logistic regression was performed to assess for factors associated with procedural success, infection, and bleeding. All demographic or procedural parameter that showed significance at the univariate analysis of Table 1 and Table 2 and BMI were included as covariates in the analysis (age, BMI, and number of truncal veins as metric covariates, sex, personal history of any previous VTE and/or SVT, previous vein treatment, CEAP classification, predominant work position as categorical covariates).

BMI showed a non-significant trend (*p* = 0.058; Exp(B) 1.166; 95%-confidence interval 0.995–1.367) to be associated with an infection. No other association with infection as well as procedural success or bleeding events was detected with *p* > 0.15 for all covariates.

### 3.5. Secondary Endpoints

Pain on the first day of postintervention was very low at 1.4 ± 1.3 on the VAS. The average concomitant analgesic intake was 2.7 ± 2.4 days. The subjective pain intensity remained low at the first (0.8 ± 1.3) and sixth week (0.4 ± 1.0) follow-up examination. At day one and week one after the intervention, there was no statistically significant trend towards increased post-interventional pain in the underweight or overweight study subpopulations. At the six-week follow-up, patients with a BMI > 30 had significantly more pain, with a VAS score of 0.84 ± 1.49. Additionally, the patients with BMI > 30 showed a longer duration of analgesic intake of 3.5 ± 2.8 days vs. 2.5 ± 2.3 days in the normal-weight population. 

Considering socioeconomic consequences, the number of work days lost was 3.6 ± 2.6 days. Patient satisfaction with the procedure at all follow-up sessions was excellent, with 96.2%–100% of patients being satisfied or very satisfied. Even in the highest BMI group, satisfaction rates of 100% at one day and 96.2% at one year after the intervention were observed. (Table 3).

## 4. Discussion

Endovenous thermal procedures are an integral part of truncal vein treatments and show comparable clinical success rates to stripping surgery [9]. However, in certain patient subpopulations, such as those with obesity, endovenous techniques are preferred over surgery for truncal vein treatment because open varicose vein surgery in patients with obesity can lead to wound complications such as fistulas or lymphoceles [6,10]. Nonetheless, endovenous procedures can be technically challenging, especially in patients with obesity, because of the deep vein location under the subcutaneous fat [10]. 

This study investigated the efficacy and safety outcomes of different BMI classes. Our results showed a very good response to therapy, with an almost 100% occlusion rate, even in the highest BMI category. Other studies with longer follow-up periods (>2 years) also demonstrated that BMI, in contrast to vein diameter and different device types, was not a risk factor for recanalization [11].

With regard to safety, the overall complication rate after ETA was low in the present study. The rate of post-interventional thromboembolic complications was 0.3% for DVT or pulmonary embolism, comparable to other study results with DVT rates of 0–5.7% [12]. Additionally, superficial thrombophlebitis was the most common complication, occurring in only 2.5% of all cases. The low thromboembolic complication rate could be explained by the consistent post-interventional anticoagulation therapy. At our study center, most patients received rivaroxaban 10 mg/day for 10 days (off-label); a thromboprophylaxis regimen that has been previously demonstrated to be effective and safe [13]. 

The overall rate of bleeding complications without major bleeding events was low (0.4%). In patients with a higher BMI, there was a slight tendency toward minor bleeding events; however, because of the low occurrence (only three patients), the clinical significance remains unclear.

Only post-interventional infections showed differences between the different weight classes. The number of post-interventional infections was significantly higher in patients with BMI > 30 compared with that in the normal-weight study population. All infections were found near phlebectomies and were all successfully treated with a brief antibiotic treatment lasting only 7–10 days. There was no need for surgical treatment such as abscess draining. No infection at the site of the treated saphenous veins was observed. However, because the absolute numbers were also low and the patients in the group with a BMI of 18.5 to 25 did not develop any infections at all, our result remains indeterminate. Furthermore, we cannot exclude the possibility that the thrombophlebits were misdiagnosed as infections.

Several studies in children and adults suggest that obesity is associated with an increased risk of infection, but that many confounding factors (e.g., comorbidities such as diabetes or malnutrition) could lead to an overestimation of the impact of body weight itself [14]. This might also be the case in our study, as we did not control for all of these comorbidities, and they might be heterogeneously distributed between the weight groups. Furthermore, the higher infection rate in patients with a BMI > 30 could be due to the higher CEAP stage (with a potentially compromised skin barrier) rather than the adipose tissue itself. However, because patients in the BMI 25–30 group also reported a higher average CEAP stage without any infection complications, there are possible alternate explanations. It is more likely that a large increase in adipose tissue activates proinflammatory cascades and, thus, may not be the only trigger for infections, but it certainly may be a major factor. Possible pathophysiological explanations for the altered immune function in patients with obesity could be an imbalance in the production of pro- and anti-inflammatory cytokines and/or hypoxia-induced increased expression of inflammatory genes [15]. Our results suggest that there might be an influence of obesity on the infection rate after ETA, but the absolute number of infections in patients with BMI > 30 was still very low (*n* = 4; 3%) and, therefore, we would still consider endovenous procedures as a favorable alternative to open surgery in these patients [16]. Regarding subjective outcome parameters, a 2020 study showed that increasing BMI was associated with poorer treatment outcomes as measured by the newly revised clinical vein severity score and CIVIQ-20 [17]. In our study, subjective satisfaction with the treatment was very high in all BMI groups, with over 96% of patients being satisfied with or very satisfied with the procedure. It is possible that appropriate pre-interventional information, as provided in our study setting, led to realistic expectations, and could, therefore, be an important factor in increasing satisfaction.

Although a higher post-interventional pain perception after ETA procedures has been described in patients with obesity [18], we did not find a higher post-interventional pain perception on the first day or in the first week after the procedure in either the obese or underweight groups than in the normal-weight group. However, we observed a significantly longer analgesic intake time in patients with obesity, which may have led to an underestimated effect. The literature suggests a close association between obesity and pain, with mechanical/structural parameters, chemical mediators, depression, sleep, and lifestyle factors linking the two confounding variables, although the exact underlying mechanisms remain unclear [18,19]. A slightly higher perception of pain in patients with obesity could also explain their significantly higher pain level in at the six-week follow-up compared to that of the normal-weight cohort, as previously described in a cohort study [18]. However, the absolute numbers were very low, as most patients were almost pain-free after the first few days post-intervention. The outpatient setting with the possibility of immediate post-interventional mobilization, one of the major advantages of this minimally invasive procedure, could explain the rapid decongestion and low post-interventional pain intensity.

One important limitation of our study is that retrospective observational studies are prone to bias. Potential confounders were identified by comparing the demographic and procedural data of the different groups. One potential confounder could have been the clinical CEAP stage of the patients because patients with a higher BMI had, on average, a more advanced stage of venous disease before the procedure. Additionally, we were only able to examine key demographics and could not rule out the possibility that other unidentified factors may have confounded the results. However, the treatment conditions were very similar and were unlikely to have significantly influenced the results. Moreover, the value of treating veins in obese patients to improve their quality of life is sometimes questioned. However, our results indicate that patients with a BMI > 30 experienced a comparable symptoms relieve and treatment satisfaction as compared to the patients with a BMI < 30. In fact, even the patients with a BMI of 50 reported high satisfaction with the treatment and it is reasonable to state that the treatment of the truncal veins is the maybe most effective way to prevent further progression of chronic venous disease.

In summary, obese patients with a BMI > 30 reported slightly more pain and prolonged analgetic intake after ETA with phlebectomies and experienced more infections at phlebectomy sites. However, the differences were small and were deemed clinically not relevant. Successful truncal vein ablation rates and patients reported satisfaction with the intervention were similar excellent across all weight groups. Thus, in conclusion, our study results demonstrate that ETA is a feasible, effective, and safe procedure for treatment of symptomatic truncal varicose veins across all BMI groups, including patients with underweight and severe obesity.

## Figures and Tables

**Table 1 jcm-12-05438-t001:** Demographic data categorized according to the BMI group.

Patient Demographics	All *n* = 796	Group 1 BMI < 18.5 *n* = 18 (2.3)	Group 2 BMI 18.5–25 *n* = 399 (50.1)	Group 3 BMI 25–30 *n* = 247 (31.0)	Group 4 BMI > 30 *n* = 132 (16.6)	*p*-Value
Age, years	54.6 ± 15.3 (19; 87)	51.58 ± 21.6	53.13 ± 15.4	56.72 ± 15.0	54.54 ± 14.1	0.022 (2 vs. 3)
Female	612 (76.9)	16 (88.9)	330 (82.7)	162 (65.6)	104 (78.8)	0.000 (2 vs. 3) 0.045 (3 vs. 4)
Thrombophilia	10 (1.3)	0 (0)	7 (1.8)	2 (0.8)	1 (0.8)	n.s.
Peripheral artery disease	22 (2.8)	2 (11.1)	8 (2.1)	6 (2.5)	6 (4.7)	n.s.
Cancer—previously active	49 (6.2) 5 (0.6)	1 (5.6) 0 (0)	28 (7.2) 2 (0.5)	17 (7.2) 1 (0.4)	3 (2.3) 2 (1.6)	n.s.
Personal history of VTE	72 (9.0)	1 (5.6)	31 (7.8)	31 (12.6)	9 (6.8)	n.s.
Personal history of SVT	123 (15.5)	0 (0)	49 (12.3)	49 (19.8)	25 (18.9)	n.s.
Personal history of any VTE and/or SVT	169 (21.2)	1 (5.6)	69 (17.3)	70 (28.3)	29 (22)	0.005 (2 vs. 3) 0.015 (2 vs. 4)
Previous vein treatment	332 (41.7)	5 (27.8)	156 (39.1)	115 (46.6)	56 (42.4)	0.028 (2 vs. 3) 0.028 (2 vs. 4)
CEAP classification						
C2	20 (2.5)	1 (5.6)	12 (3.0)	6 (2.4)	1 (0.8)	n.s.
C3	542 (68.1)	13 (72.2)	294 (73.7)	153 (61.9)	82 (62.1)	0.010 (2 vs. 3)
C4	215 (27.0)	4 (22.2)	84 (21.0)	80 (32.4)	47 (35.6)	0.008 (2 vs. 3) 0.005 (2 vs. 4)
C5/C6	19 (2.4)	0 (0)	9 (2.3)	8 (3.2)	2 (1.5)	n.s.
Profession, *n* = 737						
Employed	415 (56.3)	9 (52.9)	217 (58.2)	122 (53.7)	67 (55.8)	n.s.
Self-employed	62 (8.4)	0 (0)	37 (9.9)	17 (7.5)	8 (6.5)	n.s.
Unemployed	39 (5.3)	2 (11.8)	18 (4.8)	9 (4.0)	10 (8.3)	n.s.
Retired	221 (30.0)	6 (35.3)	101 (27.1)	79 (34.8)	35 (29.2)	n.s.
Position during work, *n* = 690						
Predominantly (>80%) standing work position	228 (33.0)	10 (71.4)	96 (27.1)	79 (37.6)	43 (38.4)	0.002 (1 vs. 2)
Predominantly (>80%) sitting work position	183 (26.5)	2 (14.3)	102 (28.8)	48 (22.9)	31 (27.7)	n.s.
Mixed work position	279 (40.4)	2 (14.3)	156 (44.1)	83 (39.5)	38 (33.9)	n.s.

Mean ± SD (min;max) or *n* (%); VTE, venous thrombo-embolism (deep vein thrombosis and/or pulmonary embolism); SVT, superficial vein thrombosis; n.s., non-significant; significant, *p* < 0.05 subgroup difference between the indicated BMI groups.

**Table 2 jcm-12-05438-t002:** Procedure data categorized according to the BMI group.

Procedure Data	All *n* = 796	Group 1 BMI < 18.5 *n* = 18 (2.3)	Group 2 BMI 18.5–25 *n* = 399 (50.1)	Group 3 BMI 25–30 *n* = 247 (31.0)	Group 4 BMI > 30 *n* = 132 (16.6)	*p*-Value
Truncal veins treated per intervention (total *n* = 1237)	1.47 ± 0.6	1.17 ± 0.38	1.47 ± 0.60	1.46 ± 0.60	1.61 ± 0.63	0.019 (1 vs. 4)
Maximum diameter of treated truncal vein, mm						
GSV	8.0 ± 3.8 (2; 28)	8.18 ± 5.23	8.15 ± 3.82	7.78 ± 3.64	7.00 ± 3.87	n.s.
AASV	6.7 ± 3.7 (2; 22)	8.00	6.43 ± 3.59	6.25 ± 2.93	7.34 ± 3.72	n.s.
SSV	6.5 ± 3.1 (2; 23)	8.50 ± 4.95	6.41 ± 3.23	6.72 ± 3.04	6.09 ± 3.07	n.s.
Length of treated truncal vein, cm						
GSV	41.8 ± 15.1	40.80 ± 16.59	41.41 ± 13.32	39.80 ± 17.79	41.73 ± 15.47	n.s.
AASV	16.1 ± 8.2	15.00	17.23 ± 8.35	15.53 ± 7.44	17.35 ± 14.59	n.s.
SSV	19.5 ± 8.0	18.00	19.31 ± 8.43	19.80 ± 8.35	19.80 ± 7.66	n.s.
Energy applied, Joule per cm						
GSV	91.7 ± 27.5	87.31 ± 9.16	91.17 ± 28.90	94.12 ± 35.22	98.47 ± 32.73	n.s.
AASV	80.9 ± 26.4	110.00	80.00 ± 31.73	83.50 ± 26.30	82.66 ± 29.13	n.s.
SSV	78.5 ± 24.1	76.00	76.24 ± 34.91	76.87 ± 18.75	82.27 ± 18.06	n.s.
Concomitant interventions						
Phlebectomy (*n*)	791 (99.4)	18 (100)	396 (99.2)	246 (99.6)	131 (99.2)	n.s.
Length of phlebectomy (cm)	69.0 ± 35.8	58.33 ± 35.69	65.51 ± 35.19	72.70 ± 37.16	72.61 ± 34.00	n.s.
Foam sclerotherapy (*n*)	174 (21.9)	2 (11.1)	85 (21.3)	61 (24.7)	26 (20.0)	n.s.
Volume of foam sclerotherapy (mL)	4.6 ± 2.10	2.50 ± 2.12	4.29 ± 1.88	4.88 ± 2.26	4.66 ± 1.91	n.s.

Mean ± SD (min;max); GSV, great saphenous vein; AASV, anterior accessory saphenous vein; SSV, small saphenous vein; n.s., non-significant; significant, *p* < 0.05 subgroup difference between the indicated BMI groups.

**Table 3 jcm-12-05438-t003:** Outcome data categorized according to the BMI group.

Outcome	All *n* = 796	Group 1 BMI < 18.5 *n* = 18	Group 2 BMI 18.5–25 *n* = 399	Group 3 BMI 25–30 *n* = 247	Group 4 BMI > 30 *n* = 132	*p*-Value
	Complete truncal vein occlusion rate (%)					
FU@d1	99.9	100	100	100	99.9	n.s.
FU@w1	99.9	100	100	100	99.9	n.s.
FU@y1	99.5	100	99.8	99.6	97.6	n.s.
	Pain (Visual Analog Scale 0–10; 0 = no pain)					
FU@d1	1.42 ± 1.32	1.88 ± 1.65	1.37 ± 1.25	1.39 ± 1.32	1.58 ± 1.32	n.s.
FU@w1	0.78 ± 1.33	1.00 ± 1.73	0.80 ± 1.28	0.77 ± 1.22	0.80 ± 1.63	n.s.
FU@w6	0.44 ± 1.04	0.29 ± 0.76	0.38 ± 0.93	0.31 ± 0.78	0.84 ± 1.49	0.005 (2 vs. 4) 0.002 (3 vs. 4)
Analgetics intake (days)	2.74 ± 2.39	2.82 ± 2.56	2.51 ± 2.27	2.78 ± 2.36	3.54 ± 2.83	0.00 (2 vs. 4) 0.029 (3 vs. 4)
	Satisfied or very satisfied with the result of the intervention (%)					
FU@d1	100	100	100	100	100	n.s.
FU@w1	98.8	100	98.8	99.1	97.5	n.s.
FU@w6	99.0	100	98.6	100	98.3	n.s.
FU@y1	98.7	100	98.5	100	96.2	n.s.
	Complications, *n* (%)					
VTE	2 (0.3)	0 (0)	1 (0.3)	1 (0.4)	0 (0)	n.s.
SVT	20 (2.5)	0 (0)	11 (2.8)	4 (1.6)	5 (3.8)	n.s.
Infection	7 (0.9)	0 (0)	3 (0.8)	0 (0)	4 (3.0)	0.020 (2 vs. 4)
Minor bleeding	3 (0.4)	0 (0)	1 (0.3)	1 (0.4)	1 (0.8)	n.s.
Major bleeding	0 (0)	0 (0)	0 (0)	0 (0)	0 (0)	n.s.

Mean ± SD (min;max); VTE, deep vein thrombosis and pulmonary embolism; SVT, superficial vein thrombosis; FU, follow-up (@w1 = 8.4 ± 3.8 days; @w6 = 48.6 ± 33.9 days; @y1 = 12.1 ± 2.9 months); n.s., non-significant; significant, *p* < 0.05 subgroup difference between the indicated BMI groups.

## Data Availability

The data supporting this study’s findings are available from the corresponding author upon reasonable request.

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
