# Peer review of "Endovenous Thermal Ablation for Treatment of Symptomatic Saphenous Veins—Does the Body Weight Matter?"

_jcm, 2023, doi:10.3390/jcm12175438_

Round 1

Reviewer 1 Report

1. Change the title.  It is not ETA of varicose veins, it is of the saphenous veins.

2. pg 1, ln 32: Change 'varicosity' to varicose veins

3. pg 2, ln 48; you have the abbreviation ETA after RFA; it is in the wrong place

4. You only used laser, not RFA.  This should also be changed in your title to only reflect EVLA, not "thermal ablation"

5. you did concommitant phlebectomy or sclerotherapy.  that likely had more to do with the higher infection rate than ETA (unless your saphenous veins were infected, which you should state)

6. line 107.  days 1, 7, 10, 6, and one year.  Probably should be 6 months.  And why would you do US on day 7 & 10?

7.  question the value of treating veins in a patient with BMI of 50.  Treating their veins will not improve the QOL.

8.  You spend a lot of time in the discussion about infections, but you didn't state exactly what the infections were: cellulitis? abscess? septic phlebitis?  Any require surgery (?I&D)  would re-write that portion of the discussion.  Again, also the fact of concomitant phlebectomy had more to do with the infections than the ablation.

Author Response

Dear Editor  

We are pleased to submit our revised manuscript entitled “Endovenous thermal ablation for treatment of symptomatic varicose veins – does the body weight matter?”

We appreciate the positive feedback from the editor and the constructive comments of the reviewers.

Following their suggestions, we have revised our manuscript. All changes are highlighted using the track changes mode. Please find our answer point by point. 

  • Reviewer 1, Comment 1: Change the title. It is not ETA of varicose veins, it is of the saphenous veins.

Thank you very much. We have changed the title "varicose veins" to "saphenous veins".

  • Reviewer 1, Comment 2: pg 1, ln 32: Change 'varicosity' to varicose veins.

"Varicosity" was replaced by "Varicose veins" on line 32 first page of the Introduction.

  • Reviewer 1, Comment 3: pg 2, ln 48; you have the abbreviation ETA after RFA; it is in the wrongplace.

Thank you for pointing this out. We have put the abbreviation RFA in the right place-see page 2 of the Introduction, line 48.

  • Reviewer 1, Comment 4: You only used laser, not RFA. This should also be changed in your title to only reflect EVLA, not "thermal ablation

Thank you for the comment. In fact we used radiofrequency ablation in 39 cases, we added this information to the results section on p. 9: “Between September 2017 and October 2020, 846 ETA interventions (807 EVLAs and 39 RFAs)..”

  • Reviewer 1, Comment 5: you did concommitant phlebectomy or sclerotherapy. that likely had more to do with the higher infection rate than ETA (unless your saphenous veins were infected, which you should state).

All local wound infections were in the area of phlebectomy. No infection at the site of the treated saphenous veins was observed. See also response to Comment 8.

  • Reviewer 1, Comment 6: line 107. days 1, 7, 10, 6, and one year.  Probably should be 6 months.  And why would you do US on day 7 & 10?

Thank you for pointing this out. We made a typing error. Follow-ups with ultrasound examinations were carried out after 1 and 7 days, 6 weeks and 1 year. We have adjusted this accordingly in the methodology p. 6: "...duplex ultrasonography on postoperative day 1, after 7 days, after 6 weeks and 1 year.

  • Reviewer 1, Comment 7: question the value of treating veins in a patient with BMI of 50. Treating their veins will not improve the QOL.

We kindly disagree. First of all, the patient subjectively experienced an improvement in his symptoms. Futhermore, as the patient presented with an advanced stage of chronic venous insufficiency (CEAP C4b) it is reasonable to treat the underlying truncal reflux to prevent disease progression. However, we agree that this an important point which we added to the discussion section, p. 14:

“Moreover, the value of treating veins in obese patients to improve their quality of life is sometimes questioned. However, our results indicate that patients with a BMI > 30 experienced a comparable symptoms relieve and treatment satisfaction as compared to the patients with a BMI<30. In fact, even the patients with a BMI of 50 reported high satisfaction with the treatment and it is reasonable to state that the treatment of the truncal veins is the maybe most effective way to prevent further progression of chronic venous disease.”

  • Reviewer 1, Comment 8: You spend a lot of time in the discussion about infections, but you didn't state exactly what the infections were: cellulitis? abscess? septic phlebitis? Any require surgery (?I&D)  would re-write that portion of the discussion.  Again, also the fact of concomitant phlebectomy had more to do with the infections than the ablation.

Thank you for the good comment. We have made a clarification under the method page 6 as to what we recorded under infections: "Suspected infectious complications with local skin infections (erysipelas, cellulitis, abscesses) that occurred in association with the procedure and required antibiotic treatment were recorded."

We have also described this in more detail in the Results p. 11:  "..., all of which were local erysipelas-like skin infections near phlebectomies...”

Also we have clarified the Discussion in this regard, see page 13: "All infections were found near phlebectomies and were all successfully treated with a brief antibiotic treatment lasting only 7-10 days. There was no need for surgical treatment such as abscess draining. No infection at the site of the treated saphenous veins was observed. "

All of the authors have read and approved the paper. It has not been published previously nor is it being considered for publication by any other peer-reviewed journal. We thank for the reviewer his helpful comments and hope that our revised manuscript will be accepted.

Sincerely yours,

Simon Bossart, M.D., Corresponding-Author

Reviewer 2 Report

Dear Authors!

Let me congratulate your team for the very interesting study performed. The data you presented is very useful for those who manage venous patients.

I have some remarks.

Major

You announced logistic regression for assessing overweight and obesity as possible risk factor for technical success and adverse events. But you did not present the results of this analysis. Please, perform multinomial logistic regression to show whether overweight is a risk factor for technical failure, pain and infection. Your sample is enough to include several variables into model.

Without this the data are incomplete.

Minor

What did you mean by symptomatic VVs? Did you include only patients who had venous pain, fatigue etc.? If no, please, exclude this definition from the title.

Lines 85-89. There is no need to describe such details. They are widely known.

I would recommend to use another definition of major and minor bleedings. Vascular specialists are usually use major and minor definitions when discussing complications of anticoagulation. And those common definitions differ from what you use in your study. So, it’s better to call it significant or insignificant or something like that. Or, you can even skip dividing at all as just three events were registered.

Line 125. Is the ref. 8 is to the point here? By the way, the paper has already received doi.

Lines 129-130. Did you screen all the patients for DVT by DUS postoperatively? If yes at which visit? Or, you performed DUS in case of symptoms suggestive for DVT? Please, describe this.

Line 132. What kind of pain did you register? And where – thigh, calf? Pain in the next few days after ETA with phlebectomy is usually related to phlebectomy. Some weeks after pain is related mainly to ablated trunks. Please, describe pain registration in detail.

Table 2. How did you measure the lengths of phlebectomy? This should be presented in Materials and Methods section.

Please, write Conclusions more to the point outlining your findings regarding connections on weight on outcomes it impacts on. 

None

Author Response

Dear Editor  

We are pleased to submit our revised manuscript entitled “Endovenous thermal ablation for treatment of symptomatic varicose veins – does the body weight matter?”

We appreciate the positive feedback from the editor and the constructive comments of the reviewers.

Following their suggestions, we have revised our manuscript. All changes are highlighted using the track changes mode. Please find our answer point by point. 

  • Reviewer 2, Comment 1: You announced logistic regression for assessing overweight and obesity as possible risk factor for technical success and adverse events. But you did not present the results of this analysis. Please, perform multinomial logistic regression to show whether overweight is a risk factor for technical failure, pain and infection. Your sample is enough to include several variables into model. Without this the data are incomplete.

Thank you very much for your comment. We have done a multivariate logistic regression and added it accordingly in the manuscript under the results by primary safety endpoints on page 11:

"Multivariate logistic regression was performed to assess for factors associated with procedural success, infection and bleeding. All demographic or procedural parameter that showed significance at the univariate analysis of Table 1 and 2 and BMI were included as covariates in the analysis (age, BMI and number of truncal veins as metric covariates, sex, personal history of any previous VTE and/or SVT, previous vein treatment, CEAP classification, predominant work position as categorical covariates).

BMI showed a non-significant trend (p=0.058; Exp(B) 1.166; 95%-confidence intervall 0.995-1.367) to be associated with an infection. No other association with infection as well as prodecural success or bleeding events was detected with p>0.15 for all covariates."

  • Reviewer 2, Comment 2: What did you mean by symptomatic VVs? Did you include only patients who had venous pain, fatigue etc.? If no, please, exclude this definition from the title.

Thank you for the comment we have better specified the term symptomatic varicose veins under Method Preoperative evaluation Page 5: “Symptomatic varicose veins were defined as varicose veins with documented reflux according to C2 after CEAP with typical venous symptoms such as pain, tightness, fatigue of the legs, restless legs, leg cramps.”

  • Reviewer 2, Comment 3: Lines 85-89. There is no need to describe such details. They are widely known.

Thank you for the comment. We would prefer to keep the definition of reflux in the manuscript as we think that there are also readers who do not know this definition.

  • Reviewer 2, Comment 4: I would recommend to use another definition of major and minor bleedings. Vascular specialists are usually use major and minor definitions when discussing complications of anticoagulation. And those common definitions differ from what you use in your study. So, it’s better to call it significant or insignificant or something like that. Or, you can even skip dividing at all as just three events were registered.

Thank you for the comment. We previously investigated the efficacy and safety of rivaroxaban for thromboprophylaxis after ETA (Uthoff H et al. Rivaroxaban for thrombosis prophylaxis in endovenous laser ablation with and without phlebectomy. J Vasc Surg Venous Lymphat Disord. 2017), using this commonly used and widely accepted definition of bleeding events. The reported regime of rivaroxaban for thrombophrophylaxis in this setting - though off-label- has been demonstrated to be safe and effective and thus is still used in our clinical practice routinely. In our perspective bleeding events are an important safety endpoint and as it is largely unknown whether weight has an impact on bleeding events after ETA interventions using medical thrombophropylaxis. Thus we believe it is important to assess and report bleeding events with an accepted and well defined endpoint with the reassuring observation in our study that these events are very rare.

  • Reviewer 2, Comment 5: Line 125. Is the ref. 8 is to the point here? By the way, the paper has already received doi.

We wanted to cite a paper with the same definition of bleeding events. We have adapted the reference to: Uthoff H, Holtz D, Broz P, Staub D, Spinedi L. Rivaroxaban for thrombosis prophylaxis in endovenous laser ablation with and without phlebectomy. J Vasc Surg Venous Lymphatic Disord. 2017 Jul;5(4):515-523. doi: 10.1016/j.jvsv.2016.12.002.

  • Reviewer 2, Comment 6: Lines 129-130. Did you screen all the patients for DVT by DUS postoperatively? If yes at which visit? Or, you performed DUS in case of symptoms suggestive for DVT? Please, describe this.
  • Yes all patients received a comprehensive ultrasound at each visit which included the deep venous system and so also screening for DVT. We have now clarified this on page 6 of the methodology under Follow up: “All patients underwent an outpatient physical examination and duplex ultrasonography (examination of the superficial and deep venous system) on postoperative day 1, after 7 days, after 6 weeks and 1 year

  • Reviewer 2, Comment 7: Line 132. What kind of pain did you register? And where – thigh, calf? Pain in the next few days after ETA with phlebectomy is usually related to phlebectomy. Some weeks after pain is related mainly to ablated trunks. Please, describe pain registration in detail.

Thank you for the comment. In everyday practise we only assessed the patient-reported general pain using the VAS score during the follow-up visits. Unfortunately, we did not obtain a more detailed pain assessment. However, in our experience (prolonged) pain was mainly caused by residual haematoma after phlebectomy or thrombosed veins after sclerotherapy.

  • Reviewer 2, Comment 8: How did you measure the lengths of phlebectomy? This should be presented in Materials and Methods section.

We have added this on page 5 under “Preoperative evaluation”:

The phlebectomy sites were marked on the skin directly preoperative with ultrasound-assistance, the phlebectomy length was then measured using a digital curvimeter (map measurer plus, Silva, Sweden).

  • Reviewer 2, Comment 9: Please, write Conclusions more to the point outlining your findings regarding connections on weight on outcomes it impacts on.

We have adjusted our Conclusion accordingly at the end of the Discussion:

In summary, obese patients with a BMI > 30 reported slightly more pain and prolonged analgetic intake after ETA with phlebectomies and experienced more infections at phlebectomy sites. However, the differences were small and deemed clinically not relevant. Successful truncal vein ablation rates and patients reported satisfaction with the intervention were similar excellent across all weight groups. Thus, in conclusion, our study results demonstrate that ETA is a feasible, effective and safe procedure for treatment of symptomatic truncal varicose veins across all BMI groups, including patients with underweight and severe obesity.

  • Editor Comment: A duplicate check has just been done for your manuscript (Similarity:

30%) and we found some overlap sentences (even seven continuous words in one sentence) with previous works in your manuscript which is not allowed by us. So we kindly suggest you to rewrite them (especially the highlighted content in Methods section) and lower significantly the similarity index when you make the revisions.

Thank you for the feedback. The 30% match is from our last publication, as we published overlapping data there as well. We have rewritten the most matched phrases in the methodology.

All of the authors have read and approved the paper. It has not been published previously nor is it being considered for publication by any other peer-reviewed journal. We thank for the reviewer his helpful comments and hope that our revised manuscript will be accepted.

Sincerely yours,

Simon Bossart, M.D., Corresponding-Author

Round 2

Reviewer 2 Report

Dear Authors!

Thank you for addressing my remarks. I support the paper acceptance.

None